# Characterization of *Herpesviridae* Family Members, BK Virus, and Adenovirus in Children and Adolescents with Nephrotic Syndrome

**DOI:** 10.3390/v16071017

**Published:** 2024-06-25

**Authors:** Silvia Mendonça Ferreira Menoni, Lucas Lopes Leon, Rodrigo Gonçalves de Lima, Anna Cristina Gervásio de Brito Lutaif, Liliane Cury Prates, Lilian Monteiro Pereira Palma, Sandra Cecília Botelho Costa, Vera Maria Santoro Belangero, Sandra Helena Alves Bonon

**Affiliations:** 1Laboratory of Virology, School of Medical Sciences, Universidade Estadual de Campinas (UNICAMP), Campinas, São Paulo 13083-887, Brazil; smenoni@yahoo.com.br (S.M.F.M.); l.lopesleon@gmail.com (L.L.L.); rolima@unicamp.br (R.G.d.L.); costa@fcm.unicamp.br (S.C.B.C.); 2Integrated Nephrology Center Unit, Pediatric Nephrology, Department of Pediatrics, School of Medical Sciences, Universidade Estadual de Campinas (UNICAMP), Campinas, São Paulo 13083-887, Brazil; aclutaif@uol.com.br (A.C.G.d.B.L.); lilianecprates@terra.com.br (L.C.P.); lilianp@unicamp.br (L.M.P.P.); vmsbelangero@gmail.com (V.M.S.B.)

**Keywords:** nephrotic syndrome, children, viral diseases, *Betaherpesviridae*, *Adenoviridae*, *Polyomaviridae*

## Abstract

Since the significance of viral infections in children and adolescents with nephrotic syndrome (NS) is yet to be defined, this study intended to estimate the occurrence, pattern, and outcomes of some DNA viral infections in children with NS. Methods: A prospective study was conducted to determine the genome identification of the viruses Epstein-Barr (EBV), human cytomegalovirus (HCMV), human herpesvirus 6 (HHV-6 type A and type B) and 7 (HHV-7), polyomavirus (BKV), and human adenovirus (HAdV) in plasma and urine samples of pediatric patients with NS. Results: A total of 35 patients aged 1 to 18 years with NS and under immunosuppressant drugs participated in the study. Plasma and urine samples were collected at regular intervals during a median follow-up of 266 days (range 133–595), and DNA was analyzed to detect the selected DNA viruses. Eleven patients (31.4%) had active virus infections, and patterns were classified as coinfection, recurrent, and consecutive. Of these, six patients (54.5%) presented viral coinfection, six (54.5%) viral recurrence, and seven patients (63.3%) had viral consecutive infection. Ten of the eleven patients with active infection had a proteinuria relapse (91%) and eight (72.7%) were hospitalized (*p* = 0.0022). Active HCMV infection was the most frequent infection and was observed in six patients (54.5%), three of the eleven patients (27.2%) had suspected HCMV disease in the gastrointestinal tract, and one had HHV-7 coinfection. The frequency of other infections was: 9% for HHV-6, 45.5% for BKV, 27.3% for HHV-7, 18.2% for EBV, and 18.2% for HAdV. Conclusion: viral infections, especially HCMV, can be an important cause of morbidity and nephrotic syndrome relapse in children.

## 1. Introduction

First described in the 15th century, nephrotic syndrome (NS) is currently recognized as a prevalent chronic illness in childhood and is primarily attributed to one of two idiopathic diseases, namely minimal-change nephrotic syndrome (MCNS) or focal segmental glomerulosclerosis (FSGS) [1]. Idiopathic NS incidence varies between 11.5 and 16.9 per 100,000 children, influenced by factors such as ethnicity and geographical location. Research suggests that idiopathic neuropathy is a result of immune dysregulation, systemic circulating factors, or inherited structural abnormalities of the podocytes [2]. Children diagnosed with NS are prone to the most prevalent glomerular disease. As treatment relies on corticosteroids, the degree of steroid use is the main predictor. What causes NS is still unclear, but the dysregulation of immune cells and the production of circulating factors that damage the glomerular filtration barrier have been described in cases of immune origin [3]. Genetic risk is more frequently described among children with steroid-resistant diseases. For most patients who respond to steroids, prednisone is the primary treatment option; however, the condition is susceptible to recurrence, requiring the administration of alternative immunosuppressive agents. The main NS complications include infections, venous thromboembolism, and an increased acute kidney damage risk. The long-term prognosis of steroid-responsive kidney disease is good, and steroid resistance is a significant predictor of continued or advanced kidney disease. The condition is often caused by trigger events, such as upper respiratory tract infections or other infections [2]. Several complications are associated with steroid-sensitive NS, including infections and toxicity from repeated courses of corticosteroids. Reduced morbidity from severe infections has been attributed to improved socioeconomic conditions, early diagnosis, and vaccination. Nevertheless, 10 to 15% of individuals with steroid-sensitive NS require hospitalization for major infections, typically during relapses or severe immunosuppression [4]. 

Nephrotic syndrome is characterized by selective proteinuria, hypoalbuminemia, hyperlipidemia, and edema. In most cases, infections lead to relapses, requiring hospitalization, and greater likelihood of morbidity and mortality [4]. Early detection of viral infections can greatly minimize the risk of comorbidities secondary to such viruses [5]. Treatment of viral infections may require the temporary reduction or cessation of immunosuppressive agents along with potential use of specific antiviral agents, contingent upon the recipient’s status, and a meticulous monitoring of kidney allograft functions [5]. Children diagnosed with NS undergo long-term immunosuppressive therapy to manage their ailments, but adverse effects involving viral reactivation may occur, such as herpesvirus and hepatitis [6]. Hepatitis B virus is characterized by high genetic diversity and naturally occurring mutations, which may be difficult to detect clinically [7,8]. Since the 1960s, several viral infections—e.g., HCMV, human immunodeficiency virus (HIV), polyomavirus (types BKV and JCV), and hepatitis A—have been implicated in renal involvement and shown to potentially lead to acute or chronic kidney injury [2]. Many viral infections can be associated with glomerular diseases and trigger glomerulonephritis in some cases. However, no specific strain of the virus has been definitively demonstrated to cause any specific renal pathology [9,10]. Different herpesviruses—e.g., EBV, HCMV, human herpesvirus type 6 (HHV-6 A/B), and human herpesvirus type 7 (HHV-7)—are of particular interest when investigating the development of several diseases in humans due to their complicated interactions with the immune system, such as persistent expression of viral antigens and viral shedding after primary infection. Since immunosuppressive drugs are commonly used to treat children with renal diseases, these patients are considered to be at high risk of developing herpesvirus reactivation [11,12,13]. BKV, a member of the *Polyomaviridae* family, causes interstitial nephritis in immunosuppressed patients. Following primary infection, which typically occurs during childhood, BKV persists mainly in the kidneys as a latent infection. Periodic reactivation results in asymptomatic shedding in the urine, thus spreading the virus in the population [14,15,16]. Reactivation occurs more often in immunosuppressed patients than in healthy individuals. Previous studies have focused on BKV reactivation and viremia incidence among kidney transplant patients [16,17]. Viruses that are part of the Human *Adenoviridae* family can cause persistent infection in human lymphoid tissues. Each HAdV species leads to the clinical manifestation of infection [18]. HAdV can infect the conjunctiva, the upper and lower respiratory tracts, and the gastrointestinal tract. It occasionally progresses into pneumonia, acute respiratory syndrome, or disseminated infection in immunosuppressed patients [19,20]. 

This prospective study was conducted with pediatric patients with NS attending follow-up at the Integrated Nephrology Center Unit, State University of Campinas (UNICAMP). The objective was to ascertain whether DNA viruses, comprising members of the *Herpesviridae* family (EBV, HCMV, HHV-6A, HHV-6B, and HHV-7), the *Adenoviridae* family (HAdV), and the *Polyomaviridae* family (BKV), could be detected in these patients during the monitoring period, and to describe the correlation between viral infections and clinical characteristics.

## 2. Patients and Methods

Inclusion criteria: pediatric patients aged 1 to 18 years under immunosuppressant drug therapy for disease control were invited to participate in the study. Exclusion criteria: patients whose parents or legal guardians did not authorize their participation, and those with insufficient volume samples for the tests. Antiviral prophylaxis was not administered. Ganciclovir treatment was administered when the N-PCR and/or *pp65* antigenemia tests were positive for active HCMV infection. *IgG* antibody levels for HCMV and EBV were evaluated by enzyme-linked immunosorbent assay (ELISA), using Abbott Laboratories: Architect CMV IgG kit and Architect EBV VCA IgG, according to the hospital routine. Other virus serology tests are not performed in the hospital routine. Longitudinal sample collection: collections were defined according to the Pediatric Nephrology Team: they began on day 0, which was defined as the first day of monitoring or on which plasma and/or urine collection started, followed by weekly sampling for the next 3 months and then monthly collections from 6 to 9 months or when clinically necessary. The biological samples were sent to the Virology Laboratory of the School of Medical Sciences, State University of Campinas, immediately after collection and stored at −20 °C until use. Variables analyzed: patient data were obtained from medical records, including gender (male/female), age, hospitalization, previous HCMV disease, HCMV-IgG, EBV-IgG, viral symptoms, fever, gastrointestinal symptoms, neurological symptoms, respiratory symptoms, creatinine clearance, albumin, urine protein/creatinine ratio, prednisone and dose, immunosuppressant regimen, length of treatment (years), corticosteroid response and relapse [21]. INS was classified according to the Kidney Disease Improving Global Outcomes (KDIGO) guidelines [22]. Creatinine clearance analysis was normalized for children using the correction factor (K) proposed by Schwartz [23,24,25]. 

This study used the following definitions of positive active viral infections in plasma and/or urine: DNA viruses coinfection: DNA detection of two or more DNA viruses in the same sample; recurrent DNA viral infection or reinfection: any infection in a patient with partial immunity acquired from a previous infection by the same virus (i.e., when DNA detection became positive after a previous negative result) [26]; consecutive DNA viral infection: two or more positive detections of viral DNA in sequentially collected plasma and/or urine samples; positive DNA viral infections: positive N-PCR results (coinfection, recurrent or consecutive) accompanied by the presence of the following symptoms: fever of unknown origin, pneumonitis (respiratory tract), central nervous system (CNS) disorders (e.g., seizures, confusion, coma, meningitis), skin rash, leukopenia, gastrointestinal infection, hepatitis, hemorrhagic cystitis; active HCMV infection: detection of one or more *pp65* antigens by an antigenemia test with immunofluorescence in more than 5 cells and/or positive N-PCR; probable HCMV disease: detection of active HCMV infection accompanied by the following clinical HCMV signs and symptoms: pneumonia, gastrointestinal disease, hepatitis, retinitis, encephalitis/ventriculitis, nephritis, cystitis, myocarditis, pancreatitis, or other end-organ diseases [27].

Plasma and urine DNA extraction: aliquot according to the DNA Biopur kit protocol (Biometrix, Curitiba, Brazil). After extracting DNA from the samples, they were subject to PCR for the β2-microglobulin gene to ensure quality and confirm the absence of PCR inhibitors [28]; nested polymerase chain reaction (N-PCR): plasma and urine specimens were subject to a nested-PCR test (N-PCR) to detect the presence of EBV [29], HCMV [30], HHV-6-A/HHV-6-B [31], HHV-7 [32], BKV [33], and HAdV [34], based on primers already described with some minor modifications. The PCR and nested-PCR (NPCR) reactions were performed using 10 μL total volume, containing 0.5 μL of extracted DNA for PCR (and 0.5 μL of the PCR product for NPCR), 5.0 μL of GoTaq^®^ Green (Promega, Winchester, VA, USA), 0.5 μL of each primer, and 3.5 μL of ultrapure water. Electrophoresis was performed using 5.0 μL of the amplified N-PCR in a 2% agarose gel stained with Unisafe Dye (Uniscience, Osasco, Brazil), then subject to ultraviolet light for visualization of the specific DNA bands (Table 1). Sizes of the final N-PCR amplification products were 209, 167, 195, 423, 264, 149, and 956 base pairs, respectively [29,30,31,32,33,34]. All tests were performed with two negative controls and one positive control from known positive patients. Purified laboratory strain HCMV-AD169 served as a positive control. All fragments amplified by N-PCR were then sequenced and their sequence was confirmed (Figure 1). CMV antigenemia test: CMV *pp65* antigenemia was assessed using a commercial immunofluorescence kit, CMV turbo Brite (Groningen, The Netherlands). It uses monoclonal antibodies specific for *pp65*, which appears in the early stages of CMV replication. Results were expressed by the number of positive cells in 2 × 10^5^ leukocytes [35].

Statistical analysis: categorical variables are presented as frequency tables with absolute (n) and percentage (%) values, and descriptive statistics of numerical variables are presented as means and medians. The correlation between positivity and categorical variables was estimated using the Chi-square test and, when necessary, Fisher’s exact test. The level of significance was set at 5%. Statistics were calculated using the SAS System for Windows (Statistical Analysis System), version 9.4 (SAS Institute Inc., 2002–2008, Cary, NC, USA. (Fleiss, J.L., 1981) [36]. 

## 3. Results

In total, 35 children and adolescents (60% male) with NS and a median age of 7 years (range 1–17 years old) participated in the study. Of these, 14 patients were ≤5 years old (40%), 12 were ≤6–10 years old (34.3%), and 9 were ≤11–17 years old (26%). Table 2 summarizes the patients’ characteristics. We analyzed 477 biological samples during the study period (206 plasma and 271 urine samples). The mean number of collections per patient was six for plasma and eight for urine. The median sampling interval during the monitoring period was 21 days (7–56 days range) (Figure 2). Patients were monitored from 9 April 2014 to 27 June 2016. Samples were tested using N-PCR techniques for viral DNA detection for a 266 days median (133–595 days range). Of the 37 biological samples positive for viral DNA, 11 were in the plasma as follows: 5 (45.5%) for HCMV, 1 (9%) for HHV-6, 3 (27.3%) for HHV-7, 1 (9%) for BKV and 1 (9%) for HAdV. Of the 26 positive urine tests, 5 (19.2%) were positive for EBV, 8 (31%) for HCMV, 2 (7.7%) for HHV-7, 10 (38.5%) for BKV, and 1 (3.8%) for HAdV. We found no significant difference in the rate of positive results on extracted viral DNA between urine and plasma (26 vs. 11, NS). BKV-DNA was detected in ten DNA-extracted urine samples but only one DNA-extracted plasma sample (*p* = 0.0277). No significant difference in the detection of viral DNA between urine and plasma samples was observed for the other viruses. HHV-6 was detected 57 days after the start of monitoring, followed by HAdv (95 days), BKV (112 days), EBV, and HCMV (168 days each), and HHV-7 (187 days). 

Frequency analysis of positive and negative results in NS patients’ samples: HCMV was the most frequent virus and was detected in eight urine samples and five plasma samples. Table 2 shows the frequency of positivity for viral DNA infections, coinfections, consecutive, and recurrence. Comparisons between patients with active viral infection and the study variable outcomes: Table 3 shows the comparisons of the results between patients with or without the virus, which show that those with an active viral infection, viral coinfection, and consecutive viral infection had a higher hospitalization rate than negative patients (72.7% vs. 16.7%, *p* = 0.0022; 83.3% vs 17.2%, *p* = 0.0118; 100% vs. 14.3%, *p* = 0.0331, respectively) (Table 3). 

## 4. Discussion

This prospective study explored active viral infections identified in children with NS during the monitoring period. Detection of the viral genome in DNA samples extracted from plasma or urine, which were collected weekly or monthly following the laboratory monitoring protocol, described the profile of active viral infections caused by EBV, HCMV, HHV-6A/B, HHV-7, BKV, and HAdV. Studies with laboratory monitoring of active viral infections are scarce and almost restricted to solid organ or stem cell transplant recipients. Administration of one or more immunosuppressant drugs in these patients makes them a high-risk group for viral reactivation, especially by latent viruses after primary infection that can later become active and cause complications [37].

Viral infection has previously been reported as a trigger for NS initiation or recurrence, especially in patients with upper respiratory tract symptoms. In this regard, BKV and HAdV have been more frequently reported as probable triggers for disease or relapse. Associations between nephrotic syndrome and the viral DNA sequences investigated in this study can be relevant for understanding the role of these viruses in patients with NS, especially in those treated with immunosuppressants [10,37]. No study has reported a possible association between active viral infections with clinical manifestations since signs and symptoms are observed via the use of appropriate antivirals, dose reduction of immunosuppressive drugs, resistance to antibiotics, HCMV disease (pneumonia), renal damage, relapse, and hospitalization [38]. Reactivation of latent viral infection associated with severe clinical symptoms and immunosuppression is a frequent finding in immunocompromised patients. In these cases, viruses proliferate within the affected cells and are released into the extracellular compartment. As viral reactivation can lead to potentially life-threatening complications in patients with impaired immune responses, early diagnosis is paramount to enable the timely initiation of appropriate antiviral therapy [38,39,40].

BK-type polyomavirus infection occurs in 3- to 5-year-old children and remains latent in the kidneys and B lymphocytes after primary infection. It has been frequently described with virus detection in the urine (viruria). Serum virus detection has recently been shown to reflect BKV-induced nephropathy in renal transplant patients, with virus multiplication in tubular epithelia and consequent graft loss [40,41]. In our study, the BK virus positivity rate was higher in urine compared with plasma samples (3.7% vs. 0.48%), and this difference was statistically significant (*p* = 0.0277). This finding reflects the presence of BKV affecting urinary epithelial cells [42]. Approximately one-third of patients with viruria will develop BK viremia (BKV), which without intervention, can progress to BKV nephropathy (BKVN), with rates varying from 1 to 10% [43,44,45]. In 2015, Yamamoto and colleagues demonstrated the kinetics of urinary BKV excretion in children with renal diseases under immunosuppressants. Serial samples of urine were collected for real-time PCR analysis from 20 patients, and BKV DNA was detected in 35% of the samples [14].

HCMV was detected in more than half of the patients with active viral infection. Of these, six patients (50%) had HCMV disease in the gastrointestinal tract (GIT). HCMV infection is also described as a possible congenital etiological factor of NS, but this relation remains unclear. Additionally, idiopathic nephrotic syndrome may develop during HCMV infection [46]. A prospective, randomized, multicenter study called NEPHROVIR evaluated 124 patients, aged 6 months to 15 years, with a recent NS diagnosis and 196 controls. The prevalence of EBV, HCMV, HHV-6, and HHV-7 was evaluated using peripheral blood samples. EBV DNA was significantly more prevalent in patients with NS compared with controls (50.8% vs. 29.1%), contrary to our findings (18.2%) [37]. In the same study, HCMV was more frequent in NS cases compared with controls (11.3 vs. 3.6%), although at lower rates than ours (54.5%). HHV-7 was detected in 83% of the NS cases and 72% of the controls in this study, whereas positivity for HHV-7 was observed in 27.3% of the NS cases. These infections were asymptomatic but should not be disregarded, as these viruses can exacerbate some diseases caused by HCMV in renal transplant patients [37,47]. Differences in the positivity rates for these viruses were high in the study by Dossier et al. (2014) due to the use of total peripheral blood and real-time PCR (qPCR), which reflects frequencies equal to those found in IgG serology and not to reactivations and viral replications, as was our objective [37]. Developing countries tend to have a higher virus seroprevalence, including in the healthy population, compared with developed countries (95% vs. 50%) [48].

Human adenovirus (HAdv) also causes infection in both immunocompromised and immunocompetent patients, the diagnosis and treatment of which remains a challenge due to its several serotypes, this infection is often asymptomatic during infancy. In a retrospective study with children receiving organ transplants, HAdv incidence varied from 6.25% to 57% [49]. However, another study with pediatric renal transplant patients observed HAdv plasma samples in 11.7% of patients using viral DNA detection by quantitative molecular techniques. Our study found a 4% positivity rate for HAdv in patients with NS. These two study patients who were positive for HAdv had viral coinfection: one with HHV-6, and the other with HHV-7, as detected in plasma samples [50,51]. Among the patients who tested positive for HAdv, one exhibited a respiratory tract symptom and another a gastrointestinal symptom and headache. However, we cannot infer that these symptoms are exclusively due to HAdv, as both patients were hospitalized due to relapse.

Patients with frequent relapses, such as study participants who remained relapsed for most of the monitoring period (>51%), have an increased risk of developing serious infections, and prolonged use of corticosteroids contributes to this scenario. Result comparisons between the positive and negative patients showed that those with active viral infection had a higher hospitalization rate than those without (72.2% vs. 16.7%). We cannot ignore hospitalizations caused by underlying disease or other infections, such as urinary tract infections (UTI), fungal infections, and respiratory infections, as they require treatment with antibiotics [4].

This study has some limitations. First, the small study population size, and second, that other DNA viruses could be associated with NS, e.g., HBV [7,8]. Future studies with larger patient samples or longitudinal case–control studies are needed to improve the level of scientific evidence.

## 5. Conclusions

This study showed that infections caused by DNA viruses, especially HCMV, can be an important cause of morbidity and nephrotic syndrome relapse in children.

## Figures and Tables

**Figure 1 viruses-16-01017-f001:**
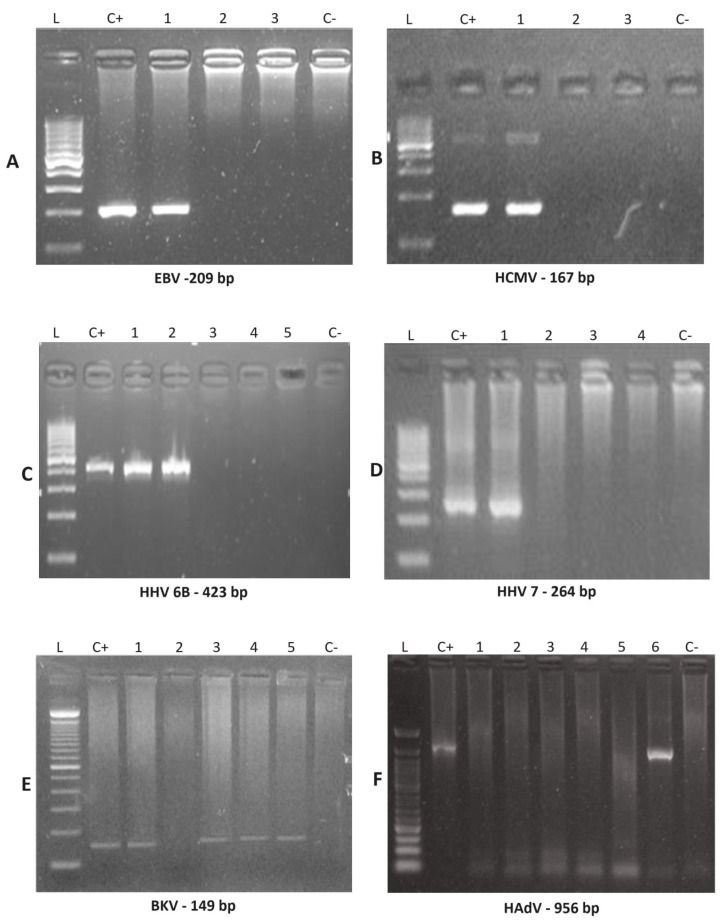
Nested PCR amplified products were examined by agarose gel electrophoresis to detect EBV, HCMV, HHV-6B, HHV-7, BKV and HAdV. Nested PCR products amplified using clinical samples (indicated by numbers) were detected for the indicated viruses by running electrophoresis on agarose gels, with 100 pb DNA size marker (L), positive control (C+) and negative control (C−). (**A**) EBV was positive for patient 21 (line #1). (**B**) HCMV was positive for patient 1 (line #1). (**C**) HHV-6B was positive for patient 24 (line #1 and #2). (**D**) HHV-7 was positive for patient 7 (Line #1). (**E**) BKV was positive for patient 8 (line #1), patient 15 (line #3), patient 21 (line #4) and patient 24 (line #5). (**F**) HAdV was positive for patient 24 (Line #6). Abbreviations: EBV, Epstein-Barr virus; HCMV, human cytomegalovirus; HHV-6A, human herpesvirus 6 type A; HHV-6B, human herpesvirus type B; HHV-7, human herpesvirus 7; BKV, human polyomavirus BKV; HAdV, human adenovirus.

**Figure 2 viruses-16-01017-f002:**
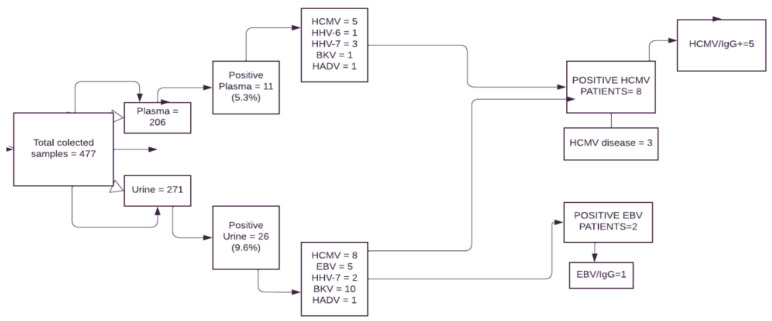
Flow chart from study participants and the results. Legend: EBV: Epstein-Barr virus; HCMV: human cytomegalovirus; HHV-6: herpesvirus type 6; HHV-7: herpesvirus type 7; HAdV: human adenovirus; BKV: polyomavirus type BK; IgG = immunoglobulin type G.

**Table 1 viruses-16-01017-t001:** Primers used in this study.

Virus	Gene	Primer	Sequence 5′—3′	PCR Condition	Base Pairs (Type)	References
EBV	EBNA-1	EBV-1 EBV-2 EBV-3 EBV-4	AAGGAGGGTGGTTTGGAAAG AGACAATGGACTCCCTTAGC ATCGTGGTCAAGGAGGTTCC ACTCAATGGTGTAAGACGAC	First round PCR: 95 °C, 2 min, followed 20 cycles 95 °C, 1 min, 55 °C, 1 min, 72 °C, 1 min Second round PCR: 95 °C, 2 min, followed 30 cycles 95 °C, 1 min, 60 °C, 1 min, 72 °C, 1 min	297 209	[27]
HCMV	HCMV UL123	HCMV-1 HCMV-2 HCMV-3 HCMV-4	ATGGAGTCCTCTGCCAAGAG CAATACACTTCATCTCCTCG GTGACCAAGGCCACGACGTT TCTGCCAGCACATCTTTCTC	95 °C, 2 min followed 30 cycles, 30 s 95 °C, 45 s, 57 °C, 1 mim, 72 °C 1 mim, 72 °C 10 min	310 167	[28]
HHV-6-A	a sequence in the BMHI K	HHV-6-1 HHV-6-2 HHV-6-3 HHV-6-4	CAAGCCCTAACTGTGTATGT TCTGCAATGTAATCAGTTTC CTGGGCGGCCCTAATAACTT ATCGCTTTCACTCTCATAAG	95 °C 3 min, followed 30 cycles 1 mim, 95 °C, 1 mim 62 °C 1 min, 72 °C 1 min, 72 °C 10 min	325 (A) 553 (B) 195 (A) 423 (B)	[29]
HHV-7	Kr4	HHV-7-1 HHV-7-2 HHV-7-3 HHV-7-4	AGTTCCAGCACTGCAATCG CACAAAGCGTCGCTATCAA CGCATACACCAACCCTACT GACTCATTATGGGGATCGAC	95 °C, 3 min, followed 30 cycles 45 s, 95 °C, 45 s, 55 °C, 1 mim, 72 °C 1 mim, 72 °C 10 min	388 264	[30]
BKV	VP2	PEP-1 PEP-2 PEP-1 BEP-1	AAGTCTTTAGGGTCTTCTAC GTGCCAACCTATGGAACAGA AAGTCTTTAGGGTCT GAGTCCTGGTGGAGT	94 °C, 5 min.94 °C, 30 s, 55 °C, 45 s 72 °C, 1 min, 72 °C, 5 min, 40 cycles	176 149	[31]
HAdV	Hexon gene sequences (adenovirus serotypes 2, 5, 40 and 41.	AdTU7′-5′ AdTU4′-5 AdnUS′-5′ AdnUA´-5	GCC ACC TTC TTC CCC ATG GC GTA GCG TTG CCG GCC GAG AA TTC CCC ATG GCN CAC AAC AC GCC TCG ATG ACG CCG CGG TG	94 °C, 1 min, 50 °C 1 min, 72 °C, 7 min, 36 Cycles	1004 956	[32]

Legend: EBV: Epstein-Barr; HCMV: human cytomegalovirus; HHV-6A: human herpesvirus 6 type A; HHV-6B: human herpesvirus type B; HHV-7: human herpesvirus 7; BKV: human polyomavirus BKV; HAdV: human adenovirus; PCR: polymerase chain reaction.

**Table 2 viruses-16-01017-t002:** Comparison of the study variables between patients with active viral infection (n = 11) and outcomes.

					Follow-Up Period							
N	Sex (M/F)/Years	HCMV/EBV IgG	Pred Group	Virus+	0–14 Days (Symp)	15–91 Days (Symp)	92–120 Days (Symp)	>121 Days (Symp)	Active Virus Infection (Type)	HCMVd	Relapse (Time %)	Hosp (Days)
1	M/8	N/A/+	2	HCMV HHV-7	0	91 91	112	140	Coinf–Rec–Cons	Y (−) *pp65*	>51%	Y (90)
7	M/14	+/+	1	HHV-7				241,266	Cons	N	>51%	Y (10)
8	M/2	+/−	1	BKV				224,252,266	Cons	N	>51%	N
11	M/14	+/−	2	HCMV				168,273,301	Rec	Y (−) *pp65*	>51%	Y (80)
15	F/11	+/+	1	HCMV BKV				448,476 342,476	Cons–Coinf	N	>51%	Y (4)
21	M/5	+/+	1	EBV HCMV BKV	7	39 42		168, 189 168	Cons–Coinf	N	>51%	Y (20)
23	F/9	−/+	1	EBV HCMV				329 329	Coinf	N	>51%	Y (26)
24	F/9	+/+	1	HHV-6 HAdV BKV	0	57 57 35			Coinf–Cons	N	>51%	Y (15)
27	F/1	+/+	1	HCMV	14	84			Rec	Y (+) HCMV/IgM-	>51%	Y (3)
31	M/7	−/+	2	HHV-7 HAdV				133–133 133	Coinf	N	N	N
34	M/3	+/+	2	BKV HCMV	0–14		112	140	Coinf–Cons	N	>51%	N

Legend: M: male; F: female; symp: symptoms; Y/N: yes/no; HAdV: human adenovirus; BKV: polyomavirus type BK; EBV: Epstein-Barr virus; HCMV: human cytomegalovirus; HHV-6: herpesvirus type 6; HHV-7: herpesvirus type 7; HCMVd: HCMV disease; Hosp: hospitalization; Coinf: coinfection; Recur: recurrent; Consec: consecutive (see definitions). Pred Group: Group 1: 0–0.5 mg/kg; Group 2: 0.51–1.0 mg/kg.

**Table 3 viruses-16-01017-t003:** Comparison of the study variables between patients with active viral infection (n = 11) vs. patients with negative results for viral coinfection, viral consecutive infection, or viral recurrent infection.

Parameter	AVI +	AVI −	*p*	Co+	Co−	*p*	(+) con	(−) con	*p*	(+) R	(−) R	*p*
Gender (Male/Female)	7/4	14/10	NS	3/3	18/11	0.6645	5/2	16/12	0.6760	3/1	18/13	0.6350
Age (1–8/9–17)	6/5	16/8	0.7077	3/3	19/10	0.6485	4/3	18/10	NS	3/1	19/22	NS
* Hospitalization (y/n)	8/3	4/20	0.0022	5/1	7/22	0.0118	5/2	7/21	0.0331	3/1	9/22	0.1061
* HCMVd (y/n)	3/8	1/23	0.0819	1/5	3/26	0.5464	1/6	3/25	NS	3/1	1/30	0.0024
Symptoms (y/n)	11/0	17/7	0.0721	6/0	22/7	0.3113	7/0	21/7	0.3009	4/0	24/7	0.5620
* Fever (y/n)	2/9	7/17	0.6855	1/5	8/21	NS	2/5	7/21	NS	1/3	8/23	NS
Gastrointestinal (y/n)	6/5	8/16	0.2831	3/3	11/18	0.6645	5/2	9/19	0.0897	3/1	11/20	0.2496
Neurological (y/n)	1/10	1/23	0.5361	0/6	2/27	NS	1/6	1/27	0.3647	1/3	1/30	0.2185
Respiratory (y/n)	7/4	11/13	0.4705	2/1	13/16	0.1774	3/4	15/13	0.6906	3/1	15/16	0.6026
Creatinine (Norm: >90)/Abn: <90)	10/1	23/1	0.5361	6/0	27/2	NS	7/0	26/2	NS	3/1	30/1	0.2185
Albumine (≤2.5/>2.51)	7/4	10/14	0.2890	5/1	12/17	0.0877	5/2	12/16	0.2285	2/2	15/16	NS
Urine prot/creat.urine (>0.20/<0.21)	4/7	11/13	0.7210	2/4	13/16	0.6804	4/3	11/17	0.4301	1/3	14/17	0.6190
Prednisone (Group 1/Group 2)	7/4	16/8	NS	4/2	19/10	NS	5/2	18/10	NS	1/3	22/9	0.1061
Immunos (Pred/Pred + Other)	3/8	7/17	NS	2/4	8/21	NS	1/6	9/19	0.6445	1/3	9/22	NS
Immunos time (1–5/6–13 years)	8/3	17/7	NS	4/2	21/8	NS	6/1	19/9	0.6445	2/2	23/8	0.5607
Corticosteroid response (S/R)	8/3	20/4	0.6524	5/1	23/6	NS	6/1	22/6	NS	2/2	26/5	0.1710
Relapse/Remission	10/1	20/4	NS	5/1	25/4	NS	7/0	23/5	0.5545	4/0	26/5	NS

Legend: p: Fisher’s test; AVI: active viral infection; co: coinfection; cons: consecutive; rec: recurrence; HCMVd: human cytomegalovirus disease; y/n: yes/no; norm: normal; abn: abnormal; Prot: protein; creat: creatinine; Immunos: immunosuppressant; Pred: prednisone; S/R: sensitive/resistant. Prednisone Group 1: 0–0.5 mg/kg; Group 2: 0.51–1.0 mg/kg. * Regarding active HCMV infection, positive patients had a higher rate of hospitalization than negative patients (*p* = 0.0006). Patients with active HCMV infection and recurrent infection had a significantly higher HCMV disease rate compared with negative patients: (50% vs. 3.6%; *p* = 0.0114) and (75% vs. 3.2%; *p* = 0.0024), respectively. Fever was significantly greater in patients positive for active HCMV infection than in HCMV-negative patients (83.3% vs. 13.8%; *p* = 0.0021). There were no differences regarding the other outcome variables studied.

## Data Availability

The protocol was designed following the requirements for research involving human subjects in Brazil and the ethical standards of the 1964 Helsinki Declaration, and it was approved by the Institutional Ethics Committee (CAAE No.: 23714313.0.0000.5404). Parents or legal guardians systematically provided written informed consent.

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
