# Peer review of "Characterization of Herpesviridae Family Members, BK Virus, and Adenovirus in Children and Adolescents with Nephrotic Syndrome"

_viruses, 2024, doi:10.3390/v16071017_

Round 1
Reviewer 1 Report (Previous Reviewer 2)
Comments and Suggestions for Authors
Comments below are only directed toward the new figures and tables.
Specific comments:
-Table 1: The EBV primer set used different PCR conditions for the first and second-round of PCR. Were the same conditions used for both first and second-round for the remaining targets?
-Figure 1: The positive (C+) and negative (C-) controls are clearly indicated, but what are the numerical samples? Not clear these are the patient evaluated in this manuscript given #2, 3, 4, 5, and 6 are not present in Table 2. Hence, it would be best to include the data from the patients that indicated in Table 2.
Editorial comments:
-Line 258: Change “Table 1” to “Table 2”.
-Table 1: Change “mim” to “min”.
Comments on the Quality of English Language
Quality of English is satisfactory. Check for minor typos.
Author Response
Ref. viruses-2969536
REVIEW 1
Comments below are only directed toward the new figures and tables.
Specific comments:
-Table 1: The EBV primer set used different PCR conditions for the first and second-round of PCR. Were the same conditions used for both first and second-round for the remaining targets?
YES.
-Figure 1: The positive (C+) and negative (C-) controls are clearly indicated, but what are the numerical samples? Not clear these are the patient evaluated in this manuscript given #2, 3, 4, 5, and 6 are not present in Table 2. Hence, it would be best to include the data from the patients that indicated in Table 2.
DONE.
Editorial comments:
-Line 258: Change “Table 1” to “Table 2”.
DONE.
-Table 1: Change “mim” to “min”.
DONE.
Thank you for your suggestions.
Reviewer 2 Report (Previous Reviewer 1)
Comments and Suggestions for Authors
In this version, the authors modified the manuscript, however, still some points missing
a) The authors mentioned that positive controls from positive hospitalized patients. The authors should know the viral load of these positive controls. Then they can determine the sensitivity and specificty of the nested PCR run in this study.
Comments on the Quality of English Language
Moderate language editing
Author Response
Ref. viruses-2969536
REVIEWER 2
In this version, the authors modified the manuscript, however, still some points missing
The authors mentioned that positive controls from positive hospitalized patients. The authors should know the viral load of these positive controls. Then they can determine the sensitivity and specificty of the nested PCR run in this study.
Unfortunately, at the time the samples were processed and the patient positive control samples were used as positive controls in the Nested PCR tests, viral load measurements of these positive controls were not performed. At the moment, we no longer have these controls, as they have already been discarded.

Round 2
Reviewer 2 Report (Previous Reviewer 1)
Comments and Suggestions for Authors
No further comments
Comments on the Quality of English Language
Fine
Author Response
25 April, 2024
REVIEW REPLY COMMENTS:
Manuscript ID: viruses-2969536
- The title of the manuscript using “DNA viruses” is inappropriate and
should be more specific. It is suggested that the title be changed to
“Characterization of Herpesviridae Family Members, BK Virus, Adenovirus in Children and Adolescents with Nephrotic Syndrome”. The title was changed.
The Introduction is out of focus and lacks organization. There are 15
paragraphs in Introduction, some of which are just one to two sentences.
Typically, Introduction contains 3-4 paragraphs covering the diseases of
interest, gaps of current research, and the rationale of the current
research. Please rearrange your Introduction to 3-4 paragraphs. Done.
3. Paragraph 8: “Children diagnosed with NS undergo long-term
immunosuppressive therapy to man-84 age their ailments, but this treatment can have adverse effects primarily involving viral reactivation, such as herpesvirus and hepatitis [6].” The authors should introduce hepatitis B virus that is characterized by high genetic diversity and naturally occurring mutations, which may lead to some difficulty in clinical detection. More references should be cited, with the following two as examples. We included these 2 references and the text in red.
Naturally occurring deletions/insertions in HBV core promoter tend to
decrease in hepatitis B e antigen-positive chronic hepatitis B patients
during antiviral therapy. Antivir Ther. 2015;20(6):623-32. doi:
10.3851/IMP2955. Epub 2015 Apr 2. PMID: 25838313.
Novel HBV recombinants between genotypes B and C in 3'-terminal reverse
transcriptase (RT) sequences are associated with enhanced viral DNA load,
higher RT point mutation rates and place of birth among Chinese patients.
Infect Genet Evol. 2018 Jan;57:26-35. doi: 10.1016/j.meegid.2017.10.023. Epub
2017 Oct 27. PMID: 29111272.
Methods: there are paragraphs containing 1-2 sentences. Please merge these paragraphs. Done
- Results: there are paragraphs containing 1-2 sentences. Please merge these paragraphs. Done
6. Discussion: there are 20 short paragraphs, with some containing 1-2
sentences. Please merge these paragraphs to 7-8 paragraphs. Done
7. There are limitations in this study that need to be discussed. First, the
small size of study population is a major flaw. Second, there are other DNA
virusesthat may be associated with NS, e.g., HBV. Done
8. Fig. 1 should add title. The legend is confusing; please indicate what are all lanes for individual viruses. Done
9. Fig. 2 please add more explanation though there is a title. Done

This manuscript is a resubmission of an earlier submission. The following is a list of the peer review reports and author responses from that submission.
Round 1
Reviewer 1 Report
Comments and Suggestions for Authors
In the second version, the authors provided serology for CMV and EBV. However, the authors did not address my previous concerns efficiently. Briefly, this paper molecular detection of DNA samples using nested PCR reactions. The authors have to show the details of the methodology used and representatives of a gel showing the expected band size of the viruses' PCR products.
The authors cite old references about nested PCR and the expected product size. The authors mentioned that PCR was performed with slight modifications. What are these modifications? what master mix was used? What are the PCR conditions and program cycles?
The authors mentioned that the sizes of the N-PCR amplification products were 209, 167, 195, 423, 264, 149 and 965. Are these sizes for EBV, HCMV, HHV-6A, HHV-6B, and HHV-7), the Adenoviridae family 76 (HAdV), and the Polyomaviridae family (BKV), respectively as ordered by the authors in the introduction?
Besides, I checked quickly some of these papers, and the product size of CMV is not correct, it should be 310 bp, not 167 which raised another suspicion about the methodology used. Also, the product size of BKV is 173 bp, not 149. These product sizes are mentioned in the papers cited by the authors.
The nested PCR for adenovirus is very big size 965 bp, I do not know why the authors used this big size, and all these papers are almost 20-30 years old. Many recent papers showed modified approaches.
Also, the authors did not provide sensitivity and specificity for the PCR product and it could not be the same as that mentioned in the original papers. Since changing lab, methodology significantly affects these values.
In addition, the sources of positive controls mentioned in the paper are not documented.
Finally, there is no rationale for performing serology for 2 out 6 or 7 tested viruses, without performing the others.
Comments on the Quality of English Language
Moderate language editing
Reviewer 2 Report
Comments and Suggestions for Authors
Human herpesviruses and BKV are ubiquitous pathogens with complex replication strategies for which latency-reactivation cycles are a continued source of viral shedding and disease burden. Menoni et al. surveyed the frequency of HCMV, EBV, HHV-6, HHV-7, BKV, and HAdv DNA detected in the plasma and urine of 35 pediatric patients with Nephrotic syndrome (NS).
Specific comments:
Lines 32-33: The data presented does not support the conclusion “Viral infections, especially HCVM, are an important cause of morbidity and nephrotic syndrome relapse in children.” Additional supporting data such as monitoring the progression or relapse over the course of antiviral therapy or animal models is required to suggests that DNA virus infection is the “cause” of morbidity and relapse in NS. It is also difficult to make this bold statement when there are no healthy patients serving as a control.
Lines 37-44: The introduction contains no background on NS which is an oversite given the target audience of this journal. Would suggest rewriting this passage into a cohesive thought process beginning with background on NS and then continue to the link with secondary viral infections.
Lines 94-97: Was there a three-month gap in sample collection between the third month and sixth? If yes, what is the reason?
Lines: 129-130: Need to provide methods for CMV immunofluorescence assay.
Lines 143-147: Questions below are in reference to the nested PCR (N-PCR) assay.
-Not explicitly stated in the N-PCR methods which DNA viruses were evaluated.
-Throughout the text “HHV-6” and “HHV-6A” is mentioned but can the assay differentiate between HHV-6A and HHV-6B?
-Given this assay is a critical feature of the manuscript it would be best to provide further details regarding the methodology (primer sequences, the number of PCR cycles, reagents, etc).
-Unclear which virus the N-PCR size products correspond too.
-What is the limit of detection (viral copies) for the nested PCR assay? How many PCR cycles were completed?
-To confirm the specificity of the assay were the positive samples sequenced? Interested in the specificity of this assay given the agarose gel was not provided.
-Given the high rate of false positives with the qualitative assay of N-PCR (assuming greater than 40 PCR cycles), viral copy numbers could have been determined with quantitative assays such as real time PCR (qPCR) or droplet digital PCR (ddPCR).
-Is it possible to differentiate a positive signal arising from the shedding of viral particles (active viral infection) vs viral DNA released from latently infected cells?
Lines 195-199: Amongst this population what is the hospitalization rate for pediatric patients with viral infection and not nephrotic syndrome? What is the hospitalization rate of pediatric patients undergoing immunosuppressive therapy relative to viral infection? Without including these necessary controls it’s not possible to conclude that “Viral infections, especially HCVM, are an important cause of morbidity and nephrotic syndrome relapse in children.”
Lines 269-272: The authors indicated the difference in positivity rates in their study compared to a previous publication (ref #33, Dossier et al.) was “due to the use of total peripheral blood.” Yes, the sample source (plasma vs total peripheral blood) may be a contributing factor, however it’s important to consider Dossier et al. used quantitative PCR (qPCR) which is a more sensitive and specific assay than N-PCR.
Lines 303-312: The omission of healthy controls and prevalence of viral DNA in other pediatric populations (immunosuppressed) hampers the ability of the authors to support their conclusions. These essential controls are necessary to provide context and significance for the limited number of patients in this study.
Lines 310-312, Table 1-2: Unclear when nephrotic syndrome relapse occurred relative to the positive viral DNA signal in the urine/plasma. Its possible that viral DNA signal could have occurred after nephrotic syndrome relapse as the virus is reactivating. As a result, its difficult to conclude that “…infections caused by DNA viruses…showed and important cause of…nephrotic syndrome relapse in children.”
Comments below are in reference to Table #1.
-Is “Pred dose” the number of times a patient was given prednisone?
-Not clear what the numbers are in reference to in columns “15-90 days”, 91-120 days”, “>121 days”. Providing a table legend with further description is suggested.
- For the “Relapse” column how was this determined and why is the duration >51% for 10 of the total 11 patients? Were the patients in relapse >51% of the study duration?
Editorial comments:
Lines 36-38; 327: Reference #1 is missing from the body of the text.
Lines 37-38: Would suggest rewording this sentence. Not completely clear what is the purpose.
Line 335: The publication date and spelling of title for this reference is incorrect.
Lines 49-52: Need to be attentive when paraphrasing other published text. The text as written in this manuscript is very similar to how its written by authors of reference #6 (Wenderfer 2015).
Example 1
This manuscript: “Viral infections are also often implicated as probable triggers for autoimmune diseases…”
Reference #6: “Viral syndromes are often implicated as probable triggers for autoimmune diseases.”
Example 2
This manuscript: “Although no virus has been definitively shown to cause any specific renal pathology…”
Reference #6: “Although no single strain of virus has been definitively shown to cause any specific renal pathology…”
Line 72: Delete extra periods.
Lines 81-82: Based on how the text is written it’s not clear if these patients have nephrotic syndrome. Patients are referred to as disease.
Lines 91-92: Details of company source and product number for ELISA is omitted.
Line: 124: Change “confection” to “coinfection”.
Line 243: Change “viruria” to “viremia”.
Table 1: Change “synt” to “symp”.
Comments on the Quality of English Language
Authors should consider editing the overall text for clarity, provide a context for nephrotic syndrome in the introduction section, and be attentive when paraphrasing publications to avoid plagiarism. See editorial comments for specific examples.